# Characterization of an IDH1 R132H Rabbit Monoclonal Antibody, MRQ-67, and Its Applications in the Identification of Diffuse Gliomas

**DOI:** 10.3390/antib12010014

**Published:** 2023-02-06

**Authors:** Raul Copaciu, Juliet Rashidian, Joshua Lloyd, Aril Yahyabeik, Jennifer McClure, Kelsea Cummings, Qin Su

**Affiliations:** Cell Marque, MilliporeSigma, 6600 Sierra College Blvd, Rocklin, CA 95677, USA

**Keywords:** isocitrate dehydrogenase 1, IDH1 R132H mutant, immunohistochemistry, monoclonal antibody, dot immunoassay, glioma

## Abstract

The current diagnosis of diffuse glioma involves isocitrate dehydrogenase (*IDH*) mutation testing. Most IDH mutant gliomas carry a G-to-A mutation at *IDH1* position 395, resulting in the R132H mutant. R132H immunohistochemistry (IHC), therefore, is used to screen for the *IDH1* mutation. In this study, the performance of MRQ-67, a recently generated IDH1 R132H antibody, was characterized in comparison with H09, a frequently used clone. Selective binding was demonstrated by an enzyme-linked immunosorbent assay for MRQ-67 to the R132H mutant, with an affinity higher than that for H09. By Western and dot immunoassays, MRQ-67 was found to bind specifically to the IDH1 R1322H, with a higher capacity than H09. IHC testing with MRQ-67 demonstrated a positive signal in most diffuse astrocytomas (16/22), oligodendrogliomas (9/15), and secondary glioblastomas tested (3/3), but not in primary glioblastomas (0/24). While both clones demonstrated a positive signal with similar patterns and equivalent intensities, H09 exhibited a background stain more frequently. DNA sequencing on 18 samples showed the R132H mutation in all IHC positive cases (5/5), but not in negative cases (0/13). These results demonstrate that MRQ-67 is a high-affinity antibody suitable for specific detection of the IDH1 R132H mutant by IHC and with less background as compared with H09.

## 1. Introduction

Significant changes have occurred in the diagnosis and classification of diffuse glial tumors during the past decade [1,2,3,4,5]. By introducing some genetic and genomic alterations into the decision-making processes, an integrated diagnosis approach has been proposed and implemented in three steps: (1) cell lineages (astrocytic, oligodendroglial, and oligoastrocytic) and histological grades (WHO grades II to IV) as proposed by traditional histopathology observation; (2) *IDH* (isocitrate dehydrogenase) gene status; and (3) 1p/19q codeletion with or without ATRX (α-thalassemia mental retardation X-linked) loss and *TP53* mutations [1,2]. This approach has been proven to be more reproducible during the diagnostic practice and more predictive for the assessment of the patient outcomes [6,7,8].

The central step in the integrated diagnosis of diffuse gliomas involves *IDH* mutation testing. This could be achieved through sequencing if tumor samples are sufficient and qualified. However, this is often not feasible during clinical practice, owing to tumor sample limitations or resource constraints, or is not recommended for cost-effectiveness considerations [9]. In these situations, immunohistochemistry (IHC) using a mutant-specific monoclonal antibody serves as an alternative procedure. It is well established that about 90% of IDH mutant gliomas carry a G-to-A mutation at *IDH1* position 395. The codon change (CGT → CAT) results in the generation of an R132H mutant with the arginine replaced by histidine [10,11,12,13]. IHC testing for the R132H mutant, therefore, has been recommended as a screening procedure [1,2,7,14] and sequencing is performed mainly for IHC-negative and IHC-equivocal samples [8,15,16,17].

Several monoclonal antibodies, mostly derived from mouse, have been described [15,18,19,20,21]. Of them, the clone H09 showed better performance and was used most frequently as a commercialized antibody. While a promising sensitivity was obtained by the test on routine formalin-fixed, paraffin-embedded (FFPE) tissue sections [17,18,22,23,24], some limitations of H09 have been described during its applications, including its cross-reactivity with other IDH1/2 mutants [16,20,25,26,27], background stain [25,28,29,30], and frequent false negativity in FFPE frozen samples following freezing and thawing procedures [15,31]. Evidently, further studies are needed into IHC applications of this antibody in terms of specificity and limitations. In addition, some new antibodies with equivalent or better performance would be valuable.

Recently, a rabbit monoclonal antibody, MRQ-67, was generated using B-cell cloning technology. The preliminary test results showed an IHC signal comparable to H09 but with less background [29]. An expanded evaluation of MRQ-67 was conducted during this study in comparison with H09, describing its peptide-recognition spectrum by liquid- and solid-phase immunoassays and further characterizing its IHC performance in gliomas.

## 2. Materials and Methods

### 2.1. Tissue Specimens

A total of 124 cases of FFPE specimens were collected from central nervous system tumors (*n* = 86) and peri-tumorous brain tissues (*n* = 38), and used for the IHC disease tissue study. Some of them were also subjected to sequencing for *IDH1* mutation. Brain tumors were diagnosed and graded principally based on the 2016 World Health Organization (WHO) Classification [1]. They consisted of 2 pilocytic astrocytomas (grade I), 22 diffuse astrocytomas (grades II and III), 15 oligodendrogliomas (grades II and III), and 27 glioblastomas (grade IV). With the guideline of the 2016 WHO Classification, grade II and III diffuse gliomas were classified into IDH mutant and IDH wild-type (wt) tumors (astrocytomas and oligodendrogliomas), respectively, and the grade IV lesions were separated into primary (IDH wt) and secondary glioblastomas (IDH mutant) [1,2]. Some updates have been made in the 2021 WHO Classification, such that IDH mutant diffuse astrocytoma and oligodendroglioma lesions (grades 2 and 3) are classified into adult-type diffuse gliomas, the IDH wt lesions are moved to other types of diffuse gliomas, and the former secondary glioblastoma lesions are designated astrocytoma—grade 4 [3,4,5]. In order to make the data reported here more comparable to the previous results, the 2016 Classification was utilized, with a note to describe possible changes in the 2021 Classification when needed. Based on the expected applications of the antibody and the observed background stain for H09 [29], 20 cases of meningioma were also tested as a reference tumor type and 38 cases of surrounding brain tissue were examined as nonneoplastic, reference tissue groups.

In addition, 111 FFPE specimens were collected from 31 types of human tissues and tested in a normal tissue study in accordance with the guideline from United States Food and Drug Administration [32]. All tissue specimens were archival samples from patients that were not individually identifiable, as described previously [29].

### 2.2. IDH1/2 Peptides

IDH peptides were synthesized in Genscript Biotech Corp (Piscataway, NJ, USA), including those for wild-type (wt) IDH1 and IDH2 and with mutations for Arg132 in IDH1 or corresponding Arg172 in IDH2, which are encountered most frequently in diffuse gliomas [1,2,6,7]. Sequences of the peptides are listed in Table 1. For the enzyme-linked immunosorbent assay (ELISA), biotinylated peptides were used.

### 2.3. ELISA

Immunoreactions of MRQ-67 with the IDH1/2 wt and mutant peptides were measured in comparison with the mouse monoclonal antibody H09 (Cat. No. Dia-H09, Dianova, Hamburg, Germany) by ELISA, as described previously [29]. Briefly, 96-well plates were coated with a 2 µg/mL streptavidin solution (Cat. No. 189730, MilliporeSigma, Billerica, MA, USA), 50 µL per well, and incubated overnight at 4 °C. Then, peptides were immobilized on the wells by incubation with a solution containing biotinylated peptides (1 µg/mL) for 1 h at room temperature. Following blocking with 5% skim milk for 1 h, the plates were probed with MRQ-67 or H09, at concentrations ranging from 1 µg/mL to 1.0 × 10^−4^ µg/mL, for 1 h at 37 °C. After washing with phosphate-buffered saline (PBS; Cat. No. 10010031, Thermo Fisher Scientific, Waltham, MA, USA) containing 0.05% Tween 20 (Cat. No. 655204, MilliporeSigma), a horseradish peroxidase (HRP)-conjugated anti-rabbit IgG (1:5000; Cat. No. 111-035-045, Jackson ImmunoResearch Lab., West Grove, PA, USA) or an anti-mouse IgG (1:5000; Cat. No. 115-035-146, Jackson ImmunoResearch Lab.) was added to the wells, for binding to MRQ-67 and H09, respectively, and incubated for 1 h at room temperature. An enzymatic reaction was conducted by adding tetramethylbenzidine (TMB) (Cat. No. CL07, MilliporeSigma) to the wells and incubation at room temperature. The reaction was stopped by adding sulfuric acid (Cat. No. LC257404, LabChem Inc., Chula Vista, CA, USA). The optical density was measured at 450 nm.

### 2.4. Dot Immunoassay

The solid-phase immuno-conjugation of MRQ-67 and H09 with IDH1/2 wt and frequently observed mutants were tested by a dot immunoassay, using a protocol as described previously [33]. Further, 2 µL of the synthetic peptides, with concentrations ranging from 1 µg/mL to 1.6 × 10^−3^ µg/mL, was dotted to the polyvinylidene difluoride (PVDF) membranes, which were supplied in the Western Breeze Chemiluminescent Kit (Cat. No. WB7104 and WB7105, Thermo Fisher Scientific). The dotted membranes were dried and blocked in the blocking solution, as provided in the kit, for 2 h at room temperature. Following incubation with MRQ-67 or H09 (0.3 µg/mL, for both) at room temperature for 1 h, the membranes were washed in the wash buffer and incubated for 1 h at room temperature with the alkaline phosphatase (AP)-conjugated anti-rabbit or anti-mouse IgG antibody, as supplied in the kits. Then, the immunoreactions were visualized by incubating the membranes in a substrate solution containing 5-bromo-4-chloro-3-indolyl phosphate (BCIP) and nitro blue tetrazolium (NBT).

### 2.5. Cell Lines and Western Blotting

A glioma cell line, BT142 mut/-, was obtained from American Type Culture Collection (ATCC; Cat. No. ACS-1018, Manassas, VA, USA), which harbors a homozygous *IDH1* R132H mutation and is used as a positive cell line for the mutant. This cell line was originally established from an anaplastic oligoastrocytoma carrying the monoallelic *IDH1* R132H mutation, but subsequently became homozygous for this mutation as a result of the loss of heterozygosity [34,35]. The cells were cultured in the NeuroCult NS-A Proliferation kit medium (Cat. No. 5751, Stem Cell Technologies, Vancouver, BC, Canada) containing 20 ng/mL recombinant human epidermal growth factor (Cat. No. 100-15, PeproTech, Cranbury, NJ, USA), 100 ng/mL recombinant human platelet-derived growth factor-AA (Cat. No. 100-13A, PeproTech), 20 ng/mL recombinant human fibroblast growth factor (Cat. No. 233-FB, R&D Systems, Minneapolis, MN, USA), and 2 µg/mL heparan sulfate (Cat. No. H3149, MilliporeSigma) and grown at 37 °C and 5% CO_2_, as proposed by the supplier. BT142 mut/- cells grow as nonadherent, sphere-forming cells in the culture medium.

HepG2, a liver epithelial cell line isolated from a hepatocellular carcinoma in a 15-year-old patient, was also obtained from ATCC (Cat. No. HB-8065). HepG2 cells have been demonstrated to carry the intact *IDH1* gene and to be positive for wt IDH1 expression [36,37]. The cells were cultured in the Eagle’s Minimum Essential Medium (EMEM; Cat. No. 30-2003, ATCC) containing 10% fetal bovine serum (Cat. No. 10438018, Thermo Fisher scientific) and grown at 37 °C and 5% CO_2_, as instructed by the supplier.

For cell lysate preparation, both cell lines were harvested and resuspended in a RIPA (radioimmunoprecipitation assay) buffer (Cat. No. 20-188, MilliporeSigma) containing a protease inhibitor cocktail (Cat. No. J61852.XF, Thermo Fisher Scientific) and lysed by an incubation at 4 °C for 15 min with agitation. The cell lysates were then centrifuged at 13,000 rpm for 20 min at 4 °C and the supernatant was collected for the Western blotting test.

For Western blotting, cell lysates were loaded onto a 4–12% Bis-Tris mini gel (Cat. No. NP0335BOX, Invitrogen, Carlsbad, CA, USA), with 50 µg of proteins in each well. Resolved proteins were electro-transferred to a PVDF membrane using an iBlot™ Stack (Cat. No. IB401002, Invitrogen). After blocking, the blots were probed with a primary antibody (MRQ-67 or H09; 1 µg/mL for both) and with an AP-conjugated anti-rabbit or anti-mouse IgG antibody, as described above for the dot blotting immunoassay. Following washing, immunoreactions were visualized by incubation in the BCIP/NBT substrate solution. A parallel reaction was performed for both cell lysates with a rabbit polyclonal IDH1 antibody (1 µg/mL; Cat. No. PA5-28206, Invitrogen) to demonstrate the presence and levels of IDH1 proteins, including the wt form and R132H mutant.

### 2.6. Cell Pellet Block Preparation

Cell pellet blocks were prepared for BT142 mut/- and HepG2 cell lines using a HistoGel procedure, based on principles as frequently used for processing cytology samples [38,39]. Briefly, the harvested cells were washed with PBS and pelleted by sedimentation. The cell pellets were then resuspended in 10% neutral buffered formalin (Cat. No. R04586-76, MilliporeSigma) and fixed for 6 h at room temperature. After incubation with 70% ethanol, the cells were suspended in the HistoGel™ Specimen Processing Gel (Cat. No. 22-110-678, Thermo Fisher Scientific) and chilled at −20 °C. Solid pellets were then placed into cassettes and processed in a TP 1020 tissue processor (Leica Biosystems, Wetzlar, Germany). Finally, the processed gel pieces were embedded in the medium Paraplast Plus^®^ (Cat. No. 39602004, Leica Biosystems).

### 2.7. Immunohistochemical Testing and Result Assessment

FFPE tissue and cell pellet sections were prepared in a thickness of 4 µm and mounted on Superfrost™ Plus slides (Fisherbrand™, Pittsburgh, PA, USA). IHC reactions were performed on the BenchMark ULTRA automated platform (Ventana Medical Systems, Inc., Tucson, AZ, USA) using an UltraView Universal DAB Detection Kit (Cat. No. 760-500, Ventana). Heat-induced epitope retrieval was performed by incubation in an EDTA-based solution (CC1; Cat. No. 950-124, Ventana) for 64 min at 95 °C, followed by incubation with the primary antibody MRQ-67 at 36 °C for 32 min. The immunoreactions were demonstrated through an HRP-based multimer detection system and visualized by incubation in a solution containing 3-3′-diaminobenzidine (DAB) and hydrogen peroxide. Nuclei were visualized by counterstaining with Hematoxylin II (Cat. No. 790-2208, Ventana) for 4 min and incubation with a bluing reagent (Cat. No. 760-2037, Ventana) for 4 min.

By titration tests, the working concentration of MRQ-67 was determined to be 2.54 µg/mL for histological specimens and 0.635 µg/mL for cell pellets. For the performance evaluation of the assay with MRQ-67 on FFPE samples, a comparative analysis was performed with the mouse monoclonal antibody H09 on selected specimens (*n* = 95). As described previously [29], H09 was used primarily at 3.25 µg/mL as the working concentration. Additional efforts were made for this antibody, in a smaller series of specimens (*n* = 41), in order to see whether it is possible to remove or to reduce the observed background stain by changing its working titer to 1.63 µg/mL. Normal rabbit (5 µg/mL; EP244, Epitomics, Burlingame, CA, USA) and mouse IgG (5 µg/mL; Thermo Fisher Scientific, Waltham, MA, USA) were used as negative controls for MRQ-67 and H09, respectively.

IHC reactions were evaluated by pathologists using a light microscope. Intensities of the positive signal, localized at the cytoplasmic and nuclear compartments of neoplastic cells (Appendix A), were graded from 0 to 4 with an increment of 0.5 (Appendix A). The signals with scores of 0.5–2, 2.5–3, and 3.5–4 were considered weak, moderate, and strong, respectively. Distribution patterns of the signal were graded by proportions of positive tumor cells and described as diffuse (51–100%), focal (5–50%), and rare (<5%). Background stain was identified by comparative observation of the results obtained with MRQ-67 and H09. Non-specific staining was also graded from 0 to 4, with those not exceeding 0.5 (faint reactions) regarded as non-disturbing and acceptable, while the stains generating a score of 1 or greater were considered disturbing background. For cases with discrepancies in scores between different slide readers, consultations were performed in the pathologist team until a consensus was reached.

### 2.8. IDH1 Genotyping

Of the neoplastic brain tissues, 34 cases were selected for DNA extraction and genotyping based on sample sizes and amounts of neoplastic cells. For comparison, 13 non-neoplastic, peri-tumorous brain tissue samples were also genotyped. Genomic DNA was extracted from FFPE tissue sections using a Deparaffinization Solution (Cat. No. 19093, Qiagen, Germantown, MD, USA) and a QIAamp DNA FFPE Tissue Kit (Cat. No. 56404, Qiagen). DNA concentrations were determined using a Nanodrop 2000 spectrophotometer (Thermo Fisher Scientific).

Genomic DNA samples were amplified by polymerase chain reaction (PCR) with a primer set designed for sanger sequencing detection of the IDH1 R132H mutation: 5′-GTG GAA ATC ACC AAA TGG CAC C-3′ (forward) and 5′-TTC ATA CCT TGC TTA ATG GGT GTA-3′ (reverse). The PCR primers were designed using NCBI Primer-BLAST tool (https://www.ncbi.nlm.nih.gov/tools/primer-blast/; accessed on 2 February 2018) to amplify a 263-base pair (bp) region of the *IDH1* gene, surrounding codon 132. Briefly, PCR reactions were performed in a 40 µL volume using 100 ng of template DNA and Phusion High-Fidelity DNA Polymerase (Cat. No. F530S, Thermo Fisher Scientific) according to the manufacturer’s instructions. PCR amplification products were verified by 2% agarose gel electrophoresis and purified using the GeneJET PCR Purification Kit (Cat. No. K0701, Thermo Fisher Scientific). Sanger DNA sequencing was performed at Genewiz (https://www.genewiz.com; accessed on 2 July 2018) using the same forward and reverse primers.

Sequence analysis was performed using the Geneious Prime sequence analysis software. Chromatogram sequence files were aligned to the IDH1 reference sequence downloaded from National Center for Biotechnology Information website (https://www.ncbi.nlm.nih.gov; accessed on 2 February 2018, Accession Number NC_000002.12:c208248510-208248248). The chromatogram sequences were inspected for the presence of a G-to-A substitution at position 395, resulting in a transition from CGT to CAT at codon 132, to determine the IDH1 genotype.

### 2.9. Statistical Analysis

The signal or background stain intensities demonstrated with different antibodies were compared using the nonparametric (Mann–Whitney) *U* test [40], and their frequencies were described with Fisher’s exact test. A *p*-value below 0.05 was considered statistically significant.

## 3. Results

### 3.1. Binding of MRQ-67 to IDH1/2 Peptides as Demonstrated by ELISA

In our previous study, ELISA was performed with the IDH1 R132H and wt peptides, and the antibody MRQ-67 was shown to be reactive with the R132H mutant peptide, but not to the wt peptide [29]. In the current observation, the liquid-phase binding analysis was expanded for MRQ-67 with IDH1/2 wt peptides and the frequently encountered mutant peptides in comparison with H09. MRQ-67 was found to bind to IDH1 mutants R132H and R132S with high affinity; to IDH1 mutants R132G, R132L, and R132C at low levels; and was not reactive to the wt IDH1/2 peptides or the three IDH2 mutant peptides tested (Figure 1A,B). For H09, a reaction was detected only to IDH1 R132H and not to the other peptides tested (Figure 1C,D). For the reaction to the IDH1 R132H mutant peptide, a higher binding affinity (>10 times) was observed with MRQ-67 than with H09. Their reaction peaks were estimated to be at a value smaller than 0.1 µg/mL and around 1 µg/mL, respectively.

### 3.2. Specific Reaction of MRQ-67 to the IDH1 R132H Mutant Peptide on Blots

In the previous study, Western blotting analysis was performed with recombinant IDH1 R132H and wt proteins, and MRQ-67 was shown to be reactive with the mutant protein, but not to the wt protein [29]. In the current observation, Western blotting was performed with cell lysates. Both MRQ-67 and H09 were shown to be reactive to the IDH1 mutant protein with R132H from BT142 mut/- and not to the wt protein from HepG2 (Figure 2A,B). On dot blots, both antibodies exhibited binding to the IDH1 R132H mutant, but not to IDH1/2 wt peptides and any other mutant peptides tested (Figure 2C,D). A higher binding capacity (>20 times) was demonstrated for MRQ-67 in comparison with H09, with a clear signal detected at the antigen levels of 8 ng/mL and 200 ng/mL, respectively.

### 3.3. IHC Binding of MRQ-67 on the IDH1 R132H Mutant Tumor Cells in Cell Pellet Blocks

IHC staining was performed comparatively on cell pellet preparations of the cell lines BT142 mut/- and HepG2. A moderate to strong immunoreactivity was observed with both MRQ-67 (0.635 µg/mL) and H09 (1.63 µg/mL) in the BT142 mut/- cell pellet, in a cytoplasmic and nuclear pattern (Figure 3A,B), while the signal was not detected in the HepG2 cells (Figure 3C,D).

### 3.4. IHC Binding of MRQ-67 in Diffuse Gliomas

In the previous study, IHC testing was performed with MRQ-67 on 32 cases of diffuse gliomas [29]. In the present study, the case number was expanded to 64, with more reference tissues included, including 20 meningiomas and 38 surrounding brain tissues. As listed in Table 2, IHC with MRQ-67 demonstrated a positive signal in a majority of diffuse gliomas including diffuse astrocytoma (16/22, 73%; Figure 4A) and oligodendroglioma lesions (grades II and III; 9/15, 60%; Figure 4C and Figure 5A). For glioblastoma, all secondary lesions were stained positive (3/3; Figure 4E), including the grade IV and grade II/III components, while all the primary lesions were shown to be negative (0/24). No positive cells were found in other samples including pilocytic astrocytomas (0/2), meningiomas (0/20), and surrounding nonneoplastic brain tissues (0/38).

As shown in Table 3, a total of 111 samples from 31 types of normal-structured human tissues were also tested for assessment of the specificity performance of MRQ-67. The specific signal was not observed in any of these samples. A weak stain was seen in kidney tissue samples (3/3) and preferentially at some proximal tubules. This stain was localized entirely at the cytoplasm. By the staining pattern, it is distinguishable from the IDH1 R132H mutant-specific immunoreactivity, as seen in BT142 mut/- cells and some diffuse gliomas, and is considered a nonspecific reaction.

### 3.5. IHC Performance of MRQ-67 in Comparison with H09

As shown in Table 4, a performance analysis was made for MRQ-67 in 95 cases of FFPE samples in comparison with H09. In terms of intensities and distribution patterns of the positive signal, MRQ-67 exhibited similar results to H09 for all positive cases (Figure 4 and Figure 5A,B), with the signal intensities ranging from moderate to strong in most tumor cells (scores, 2.5–4; mean, 3.35; *p* > 0.05 between MRQ-67 and H09).

A diffuse staining pattern was observed in all of the positive cases, with the signal present in nearly all neoplastic cells, but not visible in non-neoplastic cells. This remained true in the tumor periphery, whether with a pushing border (Figure 5A,B) or growing in infiltrating fashion (Figure 5C). Scattered tumor cells from the surrounding brain tissues, frequently few in cell numbers, mild in cell morphology, and difficult to identify for low-grade gliomas, could also be detected if they were from an IDH1 R132H-positive tumor (Figure 5D).

Background stain was evaluated for the antibody to further describe its IHC performance in comparison with H09 (Table 4, Figure 6). MRQ-67 did not exhibit a disturbing nonspecific reaction in any of the 95 samples evaluated (Figure 7A,C), with only some faint background stain observed in a minority of the samples (10/95, 11%; Figure 6A). On the contrary, H09 showed more background stain, when used at 3.25 µg/mL, in both intensities (range of scores, 0–2.5; mean, 0.45; *p* < 0.01) and frequencies of the nonspecific stain (scored 0.5 or greater; 46/95, 48%; *p* < 0.01). In addition, a disturbing background stain (scored 1 or greater) was encountered in 23% (22/95) of the samples, with the frequencies being particularly high in meningioma (12/20, 60%; *p* < 0.01) and astrocytoma samples (6/15, 40%; *p* < 0.05; Figure 6B and Figure 7B,D). 

An additional test was performed for H09 by parallel immunostaining at different working concentrations on 41 cases including 10 positive and 31 negative samples. As listed in Table 5, the immunostaining with H09 (3.25 µg/mL) demonstrated a moderate to strong signal in all 10 positive samples, equivalent to the results with MRQ-67. A disturbing nonspecific or background stain (≥1 in intensities) was seen in 11 (27%) of the samples stained with H09 (3.25 µg/mL), but not in any of the samples stained with MRQ-67 (0/41). As expected, immunostaining with H09 at 1.63 µg/mL resulted in a significant reduction in the background stain, with the disturbing stain observed in only one meningioma sample (1/41, 2%; Figure 8). The titer change, however, also resulted in a significant signal intensity reduction (≥1) in 4 (40%) of the 10 positive samples.

### 3.6. Correlation between IHC and Sequencing Results in Selective Cases

Genomic DNA was extracted from 47 cases of FFPE samples, including 34 neoplastic lesions and 13 nonneoplastic brain tissue specimens, and subjected to amplification for the IDH1 sequence by PCR. Agarose gel electrophoresis of the PCR products showed a 263 bp band for 14 neoplastic and 4 nonneoplastic brain samples. As listed in Table 6 and illustrated in Figure 9, five gliomas were shown to carry the *IDH1* R132H mutation and the remaining three harbored wt *IDH1*, with the results being consistent with the IHC test results. The sequencing test was successful in six meningiomas, all demonstrating wt *IDH1* sequence. Of them, five exhibited a disturbing background stain with H09 at 3.25 µg/mL, further confirming their nonspecific nature. As expected, four nonneoplastic brain tissue samples did not show presence of the R132H mutation.

## 4. Discussion

Identification of *IDH* mutations has been used as an initial step of glioma diagnosis and classification. Of them, *IDH1* R132H mutation is most frequent in diffuse gliomas, accounting for about 90% of the IDH mutant cases [1,2,6,7]. Immunohistochemistry with IDH1 R132H mutant-specific antibodies is used as an alternative procedure for glioma classification, being particularly useful when direct sequencing cannot be implemented for economic or facility reasons or because of tissue resource limitations [2,7,9,14]. In this study, for example, genomic DNA extraction was performed in 47 samples. Of them, only 18 (38%) exhibited qualified *IDH1* DNA amplification products and resulted in successful sequencing. For the remaining cases (29/47, 62%), DNA amplification failed. For the latter cases, IHC will be the choice for screening for *IDH1/2* mutations and identification of the IDH mutant neoplasms including diffuse gliomas. The failure in DNA amplification and the following sequencing analysis may be associated with the low yield of qualified genomic DNA as extracted from FFPE specimens, which may be caused by molecule crosslink and DNA fragmentation resulting from fixation with formaldehyde, paraffin embedding, and long-term storage of blocks, as discussed by other authors [41,42,43].

For most low-grade diffuse glioma cases, IHC testing is also a valuable tool for identification of the cryptically infiltrating, R132H mutant-positive tumor cells in the apparently uninvolved brain tissues such as surgical margins [28,29,44,45]. It may also help in identification of low-grade diffuse gliomas from reactive gliosis lesions [13,44,46,47].

Several mouse monoclonal antibodies were described as IDH1 R132H mutant-specific, including H09 [15,18], IMab-1 [19], HMab-1 [48,49], HMab-2 [50], and IHC132 [21]. Among them, H09 exhibited a more favorable performance for IHC on FFPE samples in comparison with other clones [15], and is used most frequently. While promising sensitivity and specificity have been demonstrated for the test in comparison with DNA-based sequencing in several laboratories [17,18,22,23,51], some shortcomings about H09 were mentioned in some more recent studies for its applications [16,20,26,31,52]. Its cross-reactivities with some other IDH1/2 mutants, such as IDH1 R132L and R132M, were observed, albeit at low frequencies [16,20,26,52]. Frequent false negativity and background stain were also noticed in some FFPE frozen samples following freezing/thawing procedures [31].

Recently, a rabbit monoclonal antibody, MRQ-67, was generated. A preliminary study showed its binding to IDH1 R132 mutant rather than the wt protein [29]. In the present study, the antibody was further characterized in comparison with H09. By ELISA, the antibody was found to bind to IDH1 R132H and R132S at high levels; to R132G, R132L, and R132C at low levels; and not to IDH1/2 wt peptides and other mutant peptides. By dot immunoassay, however, MRQ-67 exhibited immunoreactivity only to IDH1 R132H, but not to any other IDH1/2 mutant and wt peptides, with the results demonstrating its specificity. The differences between the liquid- and solid-phase reactions may reflect some conformational changes for the peptides following immobilization on the blots. In addition, the dot blotting results also demonstrated a higher binding affinity (about 20 times) for MRQ-67 in comparison with that for H09, in accordance with the ELISA results (>10 times higher).

While MRQ-67 was demonstrated to be a high-affinity antibody, specifically recognizing the IDH1 R132H mutant in the blotting experiments, its IHC performance was also evaluated in comparison with H09. The rabbit antibody exhibited diffuse immunoreactivity in a majority of diffuse astrocytomas (16/22, 73%), oligodendrogliomas (9/15, 60%), and all secondary glioblastomas (3/3), with the signal being comparable to that for H09 in both intensities and distribution patterns. MRQ-67 did not show any signal in other samples including in pilocytic astrocytomas, primary glioblastomas, meningiomas, and non-neoplastic brain tissues. Sequencing analysis was successful in 18 samples including 5 with IHC positivity and 13 without positive cells with MRQ-67 and H09. All cases with a detectable G-to-A substitution at the position 395 in the *IDH1* gene demonstrated positive IHC staining, whereas all cases with the confirmed wt *IDH1* sequence demonstrated negative IHC staining. These results provide further evidence for the reliability of MRQ-67 in its IHC applications. The antibody can be used as a convenient and reliable tool for classification of glial tumors and identification of diffuse astrocytoma, oligodendroglioma, and their secondary tumors. It is also valuable for distinctions of these tumors from their histological differentials including reactive gliosis, localized astrocytoma (grade I) lesions, such as pilocytic astrocytoma, primary glioblastoma, and meningioma. In addition, IDH1 R132H immunohistochemistry can be used for the demonstration of invasion fronts and margin tissue assessment, which is often a challenge during the neuropathologic practice when it is a low-grade lesion and the infiltrating cells are few.

The higher binding capacity of MRQ-67 may reflect the superiority of rabbit monoclonal antibodies over their mouse counterparts, as described in a comparative study [53]. In terms of nonspecific IHC reactions, MRQ-67 exhibited no background in most of the samples tested (92%), with some faint background stain in a minority of the samples (8%). As described in several studies, background stain and cross-reactivity is a concern for the IHC reactions with H09 [25,28,29,30,31]. This is particularly true when it is used on meningioma lesions, which tend to show some nonspecific reaction in fibrous tissue component. In this study, the background stain was observed in a substantial proportion (48%) of specimens stained with H09, when used at 3.25 µg/mL, with a disturbing background in 23% of the samples. A titration test was performed using H09 at a lower titer (1.63 µg/mL). While the titer change removed the background stain in most samples, a significant signal intensity reduction was also observed in a proportion of diffuse gliomas (40%; see Table 5). In addition, the nonspecific stain could not be removed entirely in some meningioma specimens. Special attention seems to be needed for IHC result interpretation with H09 on some tissue types like meningioma and astrocytoma. It is worth making further efforts in the future for protocol optimization with H09, and the optimal antibody concentration for IHC may be between 1.63 and 3.25 µg/mL.

In conclusion, our results have demonstrated that MRQ-67 is a high-affinity antibody with specific demonstration of the IDH1 R132H mutant by IHC and with reduced background, in some cases, as compared with clone H09. The antibody can be used for immunohistochemical identification of the IDH1 R132H mutant tumors, including diffuse gliomas.

## 5. Patents

The MRQ-67 antibody and its application to gliomas are covered by patent WO/2010/028099 and licensed by Cell Marque Corp/MilliporeSigma in North America.

## Figures and Tables

**Figure 1 antibodies-12-00014-f001:**
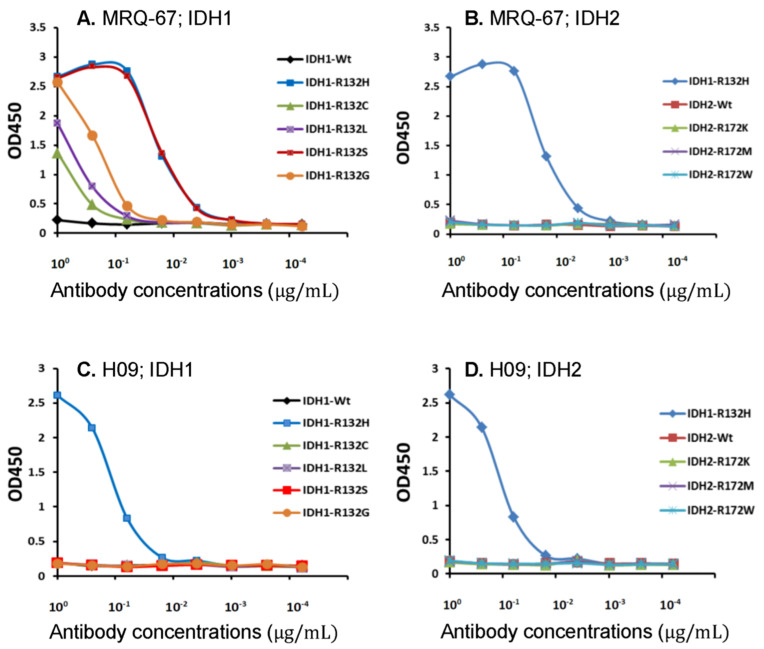
Liquid-phase binding of the antibodies MRQ-67 (**A**,**B**) and H09 (**C**,**D**) to wild-type (wt) and mutant IDH1 (**A**,**C**) and IDH2 peptides (**B**,**D**), as measured by ELISA and expressed as optical densities at 450 nm (OD450).

**Figure 2 antibodies-12-00014-f002:**
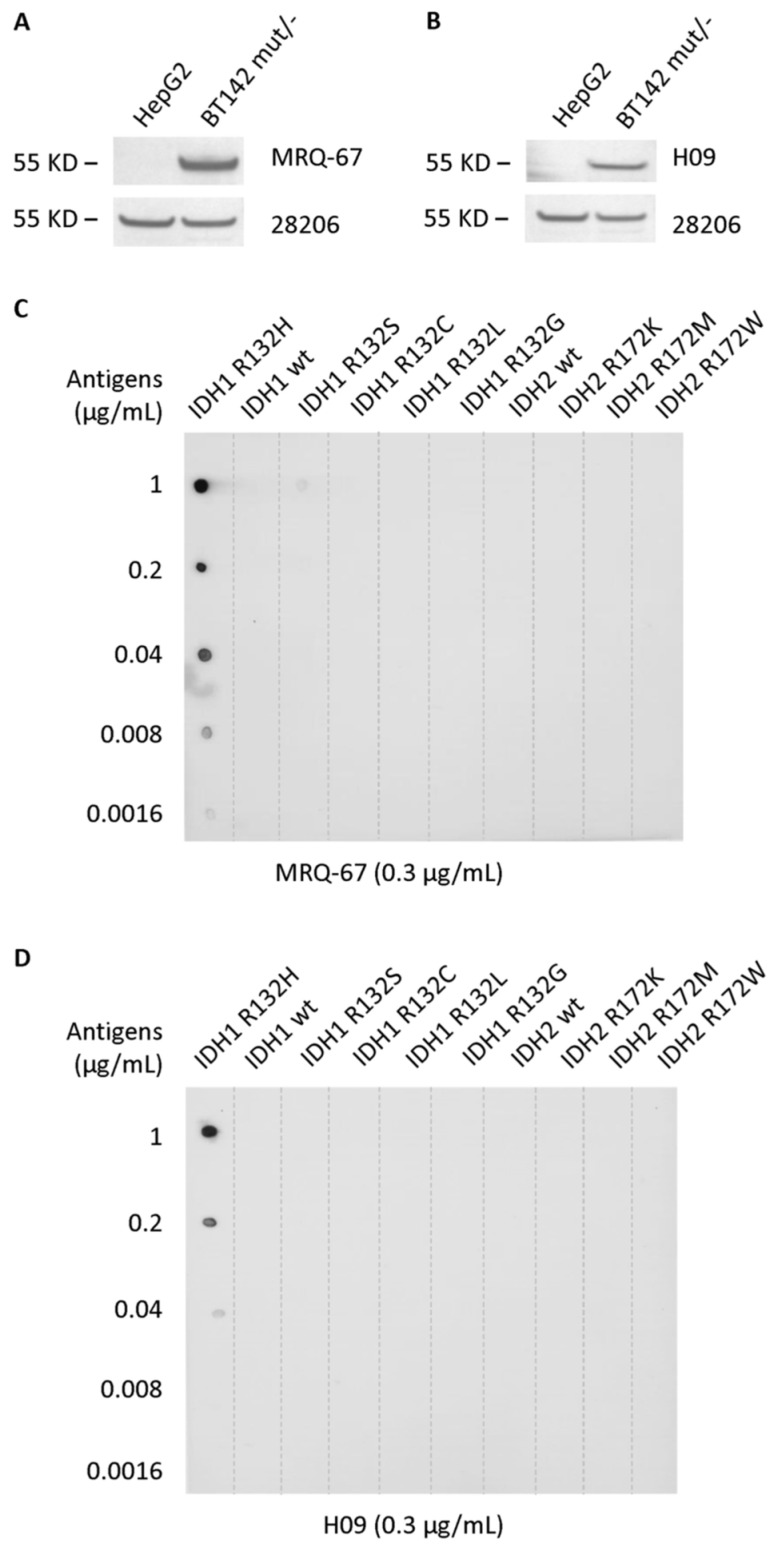
Solid-phase binding of the antibodies MRQ-67 and H09 to endogenous IDH1 R132H mutant protein on Western blots (**A**,**B**) and the synthetic R132H mutant peptide on dot blots (**C**,**D**). (**A**,**B**) Cell lysates of the glioma cell line BT142 mut/-, which harbors a homozygous IDH1 R132H mutation, and HepG2, a liver epithelial tumor cell line expressing only wt IDH1, were loaded onto a 4–12% Bis-Tris mini gel and electro-transferred onto PVDF membranes. The membranes were probed with MRQ-67 (**A**) or H09 (**B**). The presence of IDH1 protein, in both the wt and mutant forms, was confirmed by probing the membranes with a polyclonal antibody to IDH1 (28206). (**C**,**D**) Synthetic IDH1/2 wt and mutant peptides were dotted to PVDF membranes, 2 µL each, at different concentrations. The blots were probed with MRQ-67 (**C**) or H09 (**D**). Immunoreactions were demonstrated by incubation with an AP-conjugated secondary antibody and visualized in an NBT/BCIP solution.

**Figure 3 antibodies-12-00014-f003:**
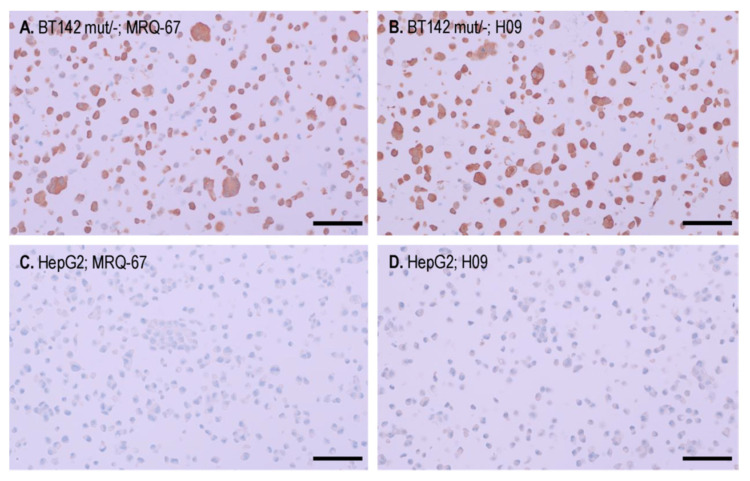
Binding of the antibodies MRQ-67 (**A**,**C**) and H09 (**B**,**D**) with endogenous IDH1 R132H mutant, but not with the IDH1 wt protein, as demonstrated on cell pellet samples of cell lines BT142 mut/- (**A**,**B**) and HepG2 (**C**,**D**) by IHC. UltraView, visualized with DAB, and counterstained with hematoxylin. Scale bar = 100 µm.

**Figure 4 antibodies-12-00014-f004:**
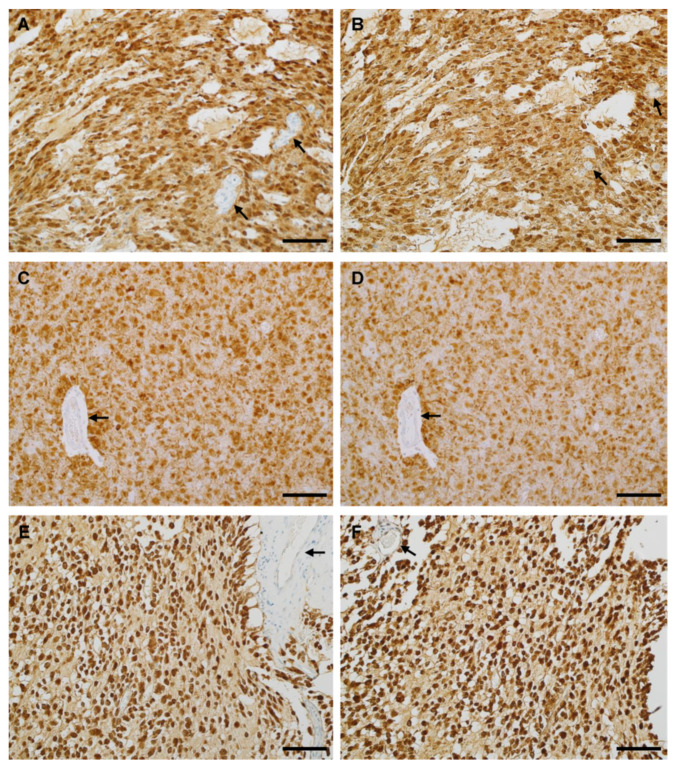
Immunoreactivity for IDH1 R132H in *IDH1*-muted gliomas including anaplastic astrocytoma (AST-19; **A**,**B**), oligodendroglioma (AST-18; **C**,**D**), and secondary glioblastoma (GLB-9; **E**,**F**) as stained separately with MRQ-67 (**A**,**C**,**E**) and H09 (**B**,**D**,**F**) on serial FFPE tissue sections. All of the tumor cells were labelled in a cytoplasmic and nuclear pattern, but blood vessels (arrow) were not. UltraView, visualized with DAB, and counterstained with hematoxylin. Scale bar = 100 µm.

**Figure 5 antibodies-12-00014-f005:**
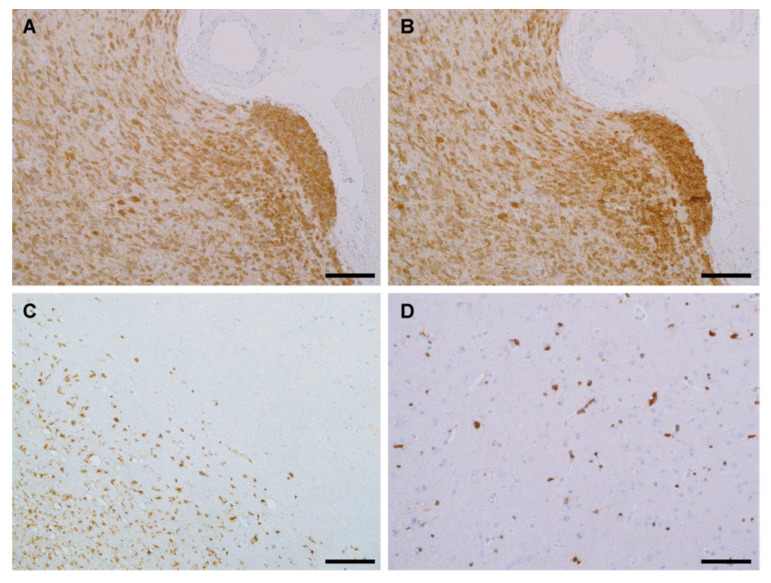
Immunohistochemical demonstration of glioma growth fronts, either with a pushing border (ODG-5; **A**,**B**) or in an infiltrating fashion (ODG-2; **C**), and cryptically infiltrating low-grade glioma cells from surrounding brain tissue (BRN-24; **D**) using MRQ-67 (**A**,**C**,**D**) or H09 (**B**). UltraView, visualized with DAB, and counterstained with hematoxylin. (**A**,**B**,**D**) scale bar = 100 µm; (**C**) scale bar = 200 µm.

**Figure 6 antibodies-12-00014-f006:**
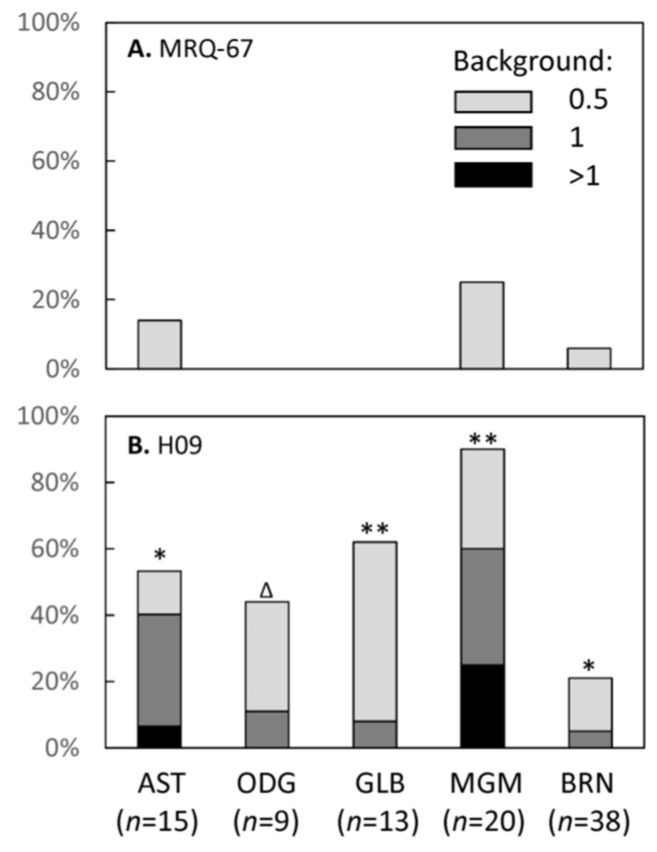
Intensities of background stain for MRQ-67 (**A**) and H09 (**B**) as encountered in astrocytoma (AST), oligodendroglioma (ODG), glioblastoma (GLB), meningioma (MGM), and surrounding nonneoplastic brain samples (BRN), with the statistical results indicated on each bar in B (*, *p* < 0.05; **, *p* < 0.01; ∆, *p* > 0.05).

**Figure 7 antibodies-12-00014-f007:**
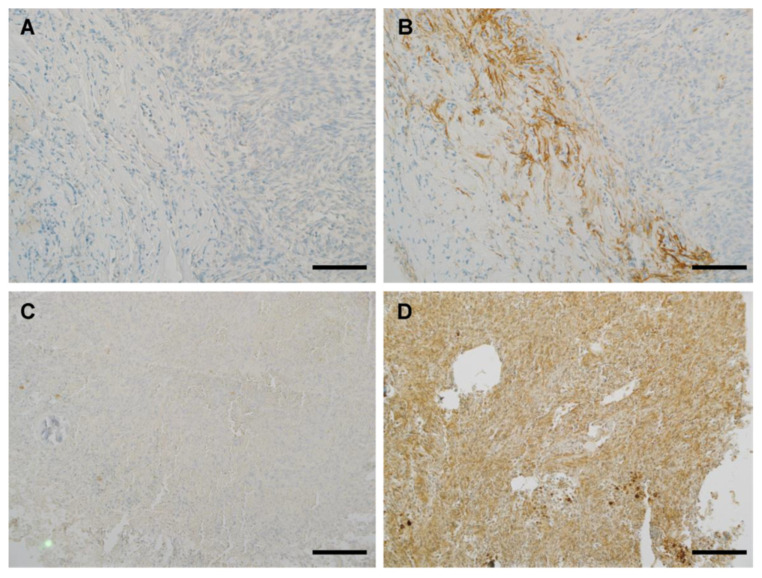
Representative meningioma samples, MGM-2 (**A**,**B**) and MGM-7 (**C**,**D**), were stained separately with MRQ-67 (**A**,**C**) and H09 (**B**,**D**), with some disturbing nonspecific stain observed with H09 (scores of 2 and 2.5 as shown in (**B**,**D**), respectively), but not with MRQ-67 (scores of 0 and 0.5 in (**A**,**C**), respectively), mainly at some fibrotic components. UltraView, visualized with DAB, and counterstained with hematoxylin. (**A**,**B**) scale bar = 100 µm; (**C**,**D**) scale bar = 200 µm.

**Figure 8 antibodies-12-00014-f008:**
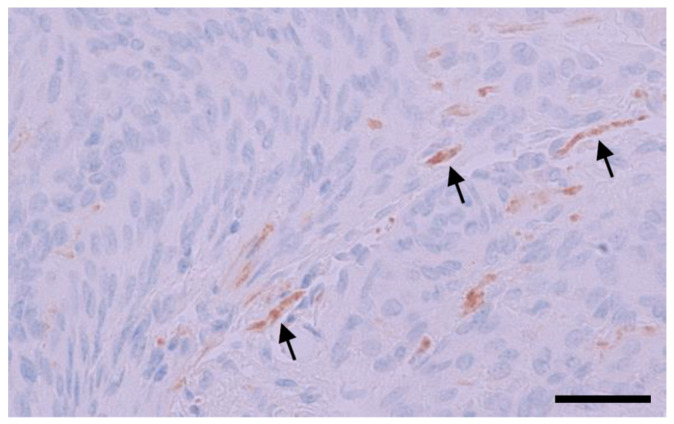
A meningioma sample (MGM-4) was stained with H09 (1.63 µg/mL), with nonspecific stain at some fibrotic components (arrows; score 1). UltraView, visualized with DAB, and counterstained with hematoxylin. Scale bar = 50 µm.

**Figure 9 antibodies-12-00014-f009:**
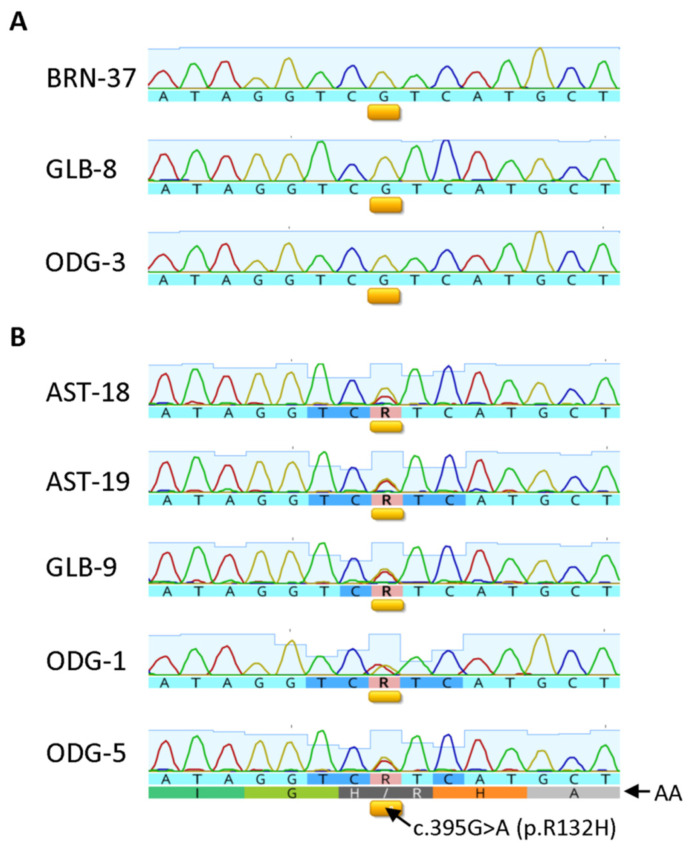
Sanger sequence chromatograms for three representative tissue samples that do not have the R132H mutation (**A**) and five tissue samples confirmed to have the R132H mutation (**B**). Corresponding amino acid sequences are labeled as AA. The c.395G > A (p.R132H) mutation location is underlined with yellow bars and labeled respectively. The presence of double chromatogram peaks corresponding to both guanine and adenine at nucleotide position c.395 indicates the presence of the c.395G > A (p.R132H) mutation and is annotated in the nucleotide sequences by the IUPAC nucleotide code R, whereas non-mutant tissue samples are identified by the presence of a single guanine peak at nucleotide position c.395 annotated in the nucleotide sequences by the code G.

**Table 1 antibodies-12-00014-t001:** IDH1/2 wild-type (wt) and mutant peptides.

Synthetic Peptides	Amino Acid Sequences
IDH1 wt	KPIIIGRHAYGD
IDH1 R132H	KPIIIGHHAYGD
IDH1 R132C	KPIIIGCHAYGD
IDH1 R132L	KPIIIGLHAYGD
IDH1 R132S	KPIIIGSHAYGD
IDH1 R132G	KPIIIGGHAYGD
IDH2 wt	KPITIGRHAHGD
IDH2 R172K	KPITIGKHAHGD
IDH2 R172M	KPITIGMHAHGD
IDH2 R172W	KPITIGWHAHGD

**Table 2 antibodies-12-00014-t002:** Immunohistochemical testing results for IDH1 R132H with MRQ-67 in 124 cases of central nervous system tumors and surrounding brain tissue.

Tumor or Tissue Types (WHO Grades)	Number of Cases Tested	Number of Cases Positive (%)
Pilocytic astrocytoma (I)	2	0 (0)
Diffuse astrocytoma (II)	19	13 (68)
Anaplastic astrocytoma (III)	3	3 (100)
Oligodendroglioma (II)	12	7 (58)
Anaplastic oligodendroglioma (III)	3	2 (67)
Primary glioblastoma (IV)	24	0 (0)
Secondary glioblastoma (IV) ^a^	3	3 (100)
Meningioma (I, II)	20	0 (0)
Nonneoplastic brain tissue	38	0 (0)

^a^ The entity was designated astrocytoma, grade 4, as proposed in the 2021 WHO Classification [3,4,5].

**Table 3 antibodies-12-00014-t003:** IHC testing results with IDH1 R132H (MRQ-67) on normal tissues.

Tissue Types	No. of Samples Tested	No. of Positive Samples
Cerebrum	11	0
Cerebellum	4	0
Adrenal gland	3	0
Ovary	3	0
Pancreas	3	0
Parathyroid	3	0
Pituitary gland	3	0
Testis	4	0
Thyroid	3	0
Breast	3	0
Spleen	3	0
Tonsil	3	0
Thymus	3	0
Bone marrow	12	0
Lung	3	0
Heart	5	0
Esophagus	4	0
Stomach	3	0
Small intestine	3	0
Colon	3	0
Liver	5	0
Salivary gland	3	0
Kidney	3	0 ^a^
Prostate	3	0
Uterus	3	0
Cervix	3	0
Skeletal muscle	3	0
Skin	3	0
Peripheral nerve	3	0
Mesothelial lining	3	0
Bladder	3	0

^a^ Weak cytoplasmic stain seen in proximal tubules in all three of the samples tested and considered a nonspecific reaction.

**Table 4 antibodies-12-00014-t004:** Performance of MRQ-67 (2.54 µg/mL) and H09 (3.25 µg/mL) in IHC on 95 FFPE tissue samples.

Tumor/Tissue Types (WHO grades)	No. of Cases Tested	No. Positive for MRQ-67 (%)	No. Positive for H09 (%)	No. with Background ≥ 1 for MRQ-67 (%)	No. with Background ≥ 1 for H09 (%)
Pilocytic astrocytoma (I)	1	0	0	0	1
Diffuse astrocytoma (II/III)	14 ^a^	10 (71)	10 (71)	0 (0)	6 (43)
Oligodendroglioma (II/III)	9 ^b^	7 (78)	7 (78)	0 (0)	1 (11)
Glioblastoma, primary (IV)	10	0 (0)	0 (0)	0 (0)	1 (10)
Glioblastoma, secondary (IV)	3	3 (100)	3 (100)	0 (0)	0 (0)
Meningioma (I/II)	20 ^c^	0 (0)	0 (0)	0 (0)	12 (60)
Non-neoplastic brain tissue	38	0 (0)	0 (0)	0 (0)	2 (5)

^a^ Including 11 diffuse astrocytomas and 3 anaplastic astrocytomas. ^b^ Including 8 oligodendrogliomas and 1 anaplastic oligodendroglioma. ^c^ Including 17 meningiomas and 3 atypical meningiomas.

**Table 5 antibodies-12-00014-t005:** Performance of the IDH1 R132H IHC in 41 FFPE samples, as demonstrated with the clone MRQ-67 or with H09 at different working concentrations.

Lesion/Tissue Types and Case Codes	Scores with MRQ-67 (2.54 µg/mL) ^a^	Scores with H09 (3.25 µg/mL) ^a^	Scores with H09 (1.63 µg/mL) ^a^
Pilocytic astrocytoma			
AST-6	0/0.5	0/1	0/0
Diffuse astrocytoma			
AST-2	3/0	3.5/0	3/0
AST-3	0/0	0/1	0/0
AST-4	3.5/0	3.5/0	3/0
AST-8	0/0	0/2	0/0
AST-16	3.5/0	3.5/0	3/0
AST-19	3.5/0	3.5/0.5	1/0 ^b^
AST-22	4/0	4/0.5	3.5/0
AST-25	3/0	3/0	2.5/0
Oligodendroglioma			
ODG-1	3.5/0	3/0	3.5/0
ODG-2	3.5/0	3.5/0.5	2.5/0 ^b^
ODG-3	0/0	0/0.5	0/0.5
ODG-6	3.5/0	3/0	1/0 ^b^
Primary glioblastoma			
GLB-2	0/0	0/0.5	0/0
GLB-3	0/0	0/0.5	0/0
GLB-5	0/0	0/0	0/0
GLB-6	0/0	0/0.5	0/0
GLB-8	0/0	0/0.5	0/0
GLB-10	0/0	0/0.5	0/0
Secondary glioblastoma			
GLB-9	4/0	4/0.5	2.5/0.5 ^b^
Meningioma			
MGM-4	0/0	0/1	0/1
MGM-9	0/0	0/0	0/0
MGM-11	0/0	0/1.5	0/0
MGM-13	0/0	0/1	0/0
MGM-14	0/0	0/0.5	0/0
MGM-16	0/0.5	0/2	0/0
MGM-17	0/0	0/2	0/0
MGM-18	0/0	0/1.5	0/0.5
MGM-19	0/0	0/1	0/0
MGM-20	0/0	0/1	0/0
MGM-21	0/0	0/0.5	0/0
Brain tissue			
BRN-20	0/0	0/0	0/0
BRN-21	0/0	0/0	0/0
BRN-25	0/0	0/0.5	0/0
BRN-31	0/0	0/0	0/0
BRN-36	0/0	0/0.5	0/0
BRN-37	0/0	0/0	0/0
BRN-41	0/0	0/0	0/0
BRN-46	0/0	0/0	0/0
BRN-53	0/0	0/0	0/0

^a^ IHC test results: scores for positive signal/scores for nonspecific or background stain. ^b^ Signal intensities reduced by a score of 1 or greater as compared with the results with H09 (3.25 µg/mL).

**Table 6 antibodies-12-00014-t006:** IDH1 R132H IHC and sequencing test results in 18 informative cases.

Case Codes	Lesion/Tissue Types	IHC Results ^a^	Sequencing Results
AST-18	Astrocytoma	Positive	R132H
AST-19	Anaplastic astrocytoma	Positive	R132H
ODG-1	Oligodendroglioma	Positive	R132H
ODG-5	Oligodendroglioma	Positive	R132H
GLB-9	Secondary glioblastoma	Positive	R132H
ODG-3	Oligodendroglioma	Negative	Wild-type
GLB-8	Primary glioblastoma	Negative	Wild-type
GLB-14	Primary glioblastoma	Negative	Wild-type
MGM-1	Meningioma	Negative	Wild-type
MGM-4	Meningioma	Negative ^b,c^	Wild-type
MGM-17	Atypical meningioma	Negative ^b^	Wild-type
MGM-18	Meningioma	Negative ^b^	Wild-type
MGM-19	Atypical meningioma	Negative ^b^	Wild-type
MGM-20	Meningioma	Negative ^b^	Wild-type
BRN-20	Cerebellar tissue	Negative	Wild-type
BRN-32	Cerebral tissue	Negative	Wild-type
BRN-37	Cerebral tissue	Negative	Wild-type
BRNCEBL-14	Cerebellar tissue	Negative	Wild-type

^a^ IHC signal was demonstrated by parallel staining with MRQ-67 and H09 (3.25 µg/mL). ^b^ Disturbing nonspecific or background stain was observed with H09 (3.25 µg/mL). ^c^ Disturbing nonspecific or background stain was observed with H09 (1.63 µg/mL).

## Data Availability

Data is contained within the article or Appendix A.

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
