# Peer review of "Characterization of an IDH1 R132H Rabbit Monoclonal Antibody, MRQ-67, and Its Applications in the Identification of Diffuse Gliomas"

_2073-4468, 2023, doi:10.3390/antib12010014_

Round 1
Reviewer 1 Report
In this manuscript, the authors present the characterization of a rabbit monoclonal antibody, MRQ-67 that detect IDH mutations. It is demonstrate here that MRQ-67 antibody is slightly better than H09, the most frequently used Ab, in terms of specificity and sensitivity. Therefore, the authors propose the usage of this novel antibody for detecting IDH1 5132H mutation in patient tumor samples.
Background, experiments and results are clearly presented, however, some points should be considered for optimizing this manuscript.
- Quantification by IHC has to be clearly defined. Grade 0 to 4 should correspond to a certain intensity and/or a % of positive cells. Each grade should be better explained and supported by a supplementary Figure showing an example of each Grade.
- Same remark for the background quantification.
- Tables 1, 2, 4 and 5 show data that should be presented in Results and not in Materials and Methods part.
- When they comment IHC pictures, the authors mention cytoplasmic and nuclear staining, that it is not observable with these Figures. Consequently, images with greater magnitude should be presented to clearly see that.
- Data with H09 in Table 5 suggest that the optimal concentration might be between the two tested concentrations. This point could be discussed by the authors.
Author Response
General comments
In this manuscript, the authors present the characterization of a rabbit monoclonal antibody, MRQ-67 that detect IDH mutations. It is demonstrate here that MRQ-67 antibody is slightly better than H09, the most frequently used Ab, in terms of specificity and sensitivity. Therefore, the authors propose the usage of this novel antibody for detecting IDH1 5132H mutation in patient tumor samples.
Background, experiments and results are clearly presented, however, some points should be considered for optimizing this manuscript.
Authors’ response: thank you!
Specific suggestions
- Quantification by IHC has to be clearly defined. Grade 0 to 4 should correspond to a certain intensity and/or a % of positive cells. Each grade should be better explained and supported by a supplementary Figure showing an example of each Grade.
Response of authors:
This is a great suggestion. For this reason, an additional figure (Supplementary Figure 1) was prepared and presented in the Section 2.7 (Lines 221-226 in the current version), in order to illustrate the result assessment. Accordingly, it is cited in the text in Lines 205-206 (current version).
Quantitative analysis for IHC signal is a very challenging topic, being a good idea but difficult to realize it. This is particularly true for low-grade diffuse gliomas where tumor cell numbers and distribution patterns are highly varying, and sometimes it is even difficult to identify the tumor cells from reactive glial cells with full confidence. For these reasons, the result assessment was performed in two steps, as adopted by most IHC studies. First, the results are assessed qualitatively as positive (scores 0.5 or greater) or negative (0). Second, for the positive samples, a semi-quantitative procedure is adopted to grade the positive immunoreactivity by scores, 0.5 to 4, or by levels, such as weak (low level), moderate (intermediate level) and strong (high level). Obviously, the IHC result assessment, as performed by most pathologists, is not perfect, but it works for most conditions.
For these reasons, background stain is identified and assessed following similar procedures. The background stain or nonspecific stain were also assessed in a semi-quantitative manner.
- Same remark for the background quantification.
Responses of authors:
This is a great proposal! We should make some additional efforts in the nonspecific, background stain assessment. For this purpose, we have updated the legend for Figures 7 and 8, with the scores included for the nonspecific stain as shown in separate parts.
Lines 378-384 (former Lines 361-364), legends of Figure 7: “Figure 7. Representative meningioma samples, MGM-2 (A,B) and MGM-7 (C,D), were stained separately with MRQ-67 (A,C) and H09 (B,D), with some disturbing nonspecific stain observed with H09 (scores 2 and 2.5 as shown in B and D, respectively), but not with MRQ-67 (scores 0 and 0.5 as shown in A and C, respectively), mainly at some fibrotic components. UltraView, visualized with DAB, and counterstained with hematoxylin. A and B, scale bar = 100 µm; C and D, scale bar = 200 µm.”
Lines 400-402 (former Lines 376-378), legend of Figure 8: “Figure 8. A meningioma sample (MGM-4) was stained with H09 (1.63 µg/mL), with nonspecific stain at some fibrotic components (arrows; score 1). UltraView, visualized with DAB, and counterstained with hematoxylin. Scale bar = 50 µm.”
- Tables 1, 2, 4 and 5 show data that should be presented in Results and not in Materials and Methods part.
Responses of authors:
This is a great suggestion! Corresponding changes have been done for Tables 1-5.
Table 3 (Line 105, in original version) has been changed to Table 1 (Line 98, current version), with a corresponding change (“Table 3” to “Table 1”) in text (currently, Line 96; originally, Line 102).
Table 1 (Lines 87-90, in original version) has been changed to Table 2 (Lines 320-323, current version), and moved from Section 2.1 to the Section 3.4. Corresponding changes were also made in text, with “Table 1” removed from original Line 67, Section 2.1 and “Table 1” changed to “Table 2” in the Section 3.4 (currently, Line 313).
Table 2 (Line 95-97, in original version) has been changed to Table 3 (Line 331-333, current version), and moved from Section 2.1 to the Section 3.4. Corresponding changes were also made in text, with “Table 2” removed from the original Line 93, Section 2.1 and “Table 2” changed to “Table 3” in the Section 3.4 (currently, Line 324).
Table 4 has been moved from Section 2.7 (Lines 209-212 in original version) to the Section 3.5 (Lines 340-343 in the current version). Corresponding changes were also made in text, with “Table 4” removed from the original Line 201, Section 2.7.
Table 5 has been moved from Section 2.7 (Lines 213-216 in original version) to the Section 3.5 (Lines 395-398 in the current version). Corresponding changes were also made in text, with “Table 5” removed from the original Line 204, Section 2.7.
- When they comment IHC pictures, the authors mention cytoplasmic and nuclear staining, that it is not observable with these Figures. Consequently, images with greater magnitude should be presented to clearly see that.
Responses of authors:
Thanks for the comments. The cytoplasmic and nuclear staining pattern is shown in the updated Figure 4, particularly in the parts A, B and E, F. The reaction is localized in cytoplasm, also with nuclear stain in many tumor cells. For a clearer explanation, the figure legend was also updated (Currently, Line 356).
For a clearer illustration, as suggested by the reviewer, we add a figure (Supplementary Figure 1; Lines 215-220) with 3 additional high-magnification photos to show the cytoplasmic and nuclear staining pattern for IDH1 R132H in both cell pellet block (A) and tumor biopsy specimens (B, C).
- Data with H09 in Table 5 suggest that the optimal concentration might be between the two tested concentrations. This point could be discussed by the authors.
Responses of authors:
This is a great suggestion! We have made the following changes in the Discussion section.
Line 500 (current version), a typo was found in the bracket, with “Table 6” changed to “Table 5”.
Lines 520-522 (current version), a brief discussion is added for the possible optimal antibody concentration for IHC with H09.
Reviewer 2 Report
The manuscript describes the characetrization of a rabbit monoclonal antibody specific to a mutant variant of the protein IDH1/IDH2 with respect to antigen affinity and selectivity in liquid and solid-phase immunoassays and its IHC performance. The motivation and study design are clearly presented.
Following remarks should be considered:
- more precise distinction from the work already published (citation #29) should be added to the results section. Some experiments and figures are very similar (Figure 4).
- if data are available showing IHC from FFPE samples following freeze/thawing in comparison to H09, it would further contribute to a better characterization of the antibody.
- Since the antibody concentration used in the dot blot (0.3 µg/ml) is lower than the concentration used for cell pellet analysis (0.64 µg/ml) and the concentration used for histological samples (2.54 µg/ml), please add a comment as to whether the cross-reactivity towards the R132S ( & R132G) mutant could be of concern for the IHC analysis.
- correct line 38: "sequencing if with sufficient..."
Author Response
General comments
The manuscript describes the characetrization of a rabbit monoclonal antibody specific to a mutant variant of the protein IDH1/IDH2 with respect to antigen affinity and selectivity in liquid and solid-phase immunoassays and its IHC performance. The motivation and study design are clearly presented.
Response of authors:
Thank you!
Specific comments and/or suggestions
- Comments and/or suggestions of the reviewer:
More precise distinction from the work already published (citation #29) should be added to the results section. Some experiments and figures are very similar (Figure 4).
Responses of authors:
This is a great suggestion! For this purpose, corresponding changes were made in the Result sections.
In our previous study, antibody generation of MRQ-67 was described and a preliminary performance test was performed [29]. In the current observation, ELISA, Western blotting and IHC characterization was expanded, with more sample types and numbers included. In addition, dot immunoassay, cell culture and sequencing analyses were also made to further describe its performance in comparison to H09.
The distinction was made precisely adding a description in all related paragraphs including the Sections 3.1 (Lines 261-265), 3.2 (Lines 277-280) and 3.4 (Lines 310-312).
We have checked the photos in Figure 4. The photos A and B look similar to our published work (citation #29, Figure 2 a and b), but they are not same. In response to your suggestions, we have updated Figure 4 (currently, Line 352), with the potentially questionable parts (A and B) replaced by two new photos which were taken from the same case but from a different tissue block. Accordingly, the legend was also updated (currently, Lines 353-357), with minimal changes in the Line 357.
- Comments and/or suggestions of the reviewer:
If data are available showing IHC from FFPE samples following freeze/thawing in comparison to H09, it would further contribute to a better characterization of the antibody.
Responses of authors:
This is a great suggestion. A comparative IHC test on FFPE samples following freeze/thawing may provide some confirmatory results as described with H09 by Yoshida et al (citation #31), and disclose whether this will happen with MRQ-67 or not. It may also result in some mechanistic understanding for the changes in the IDH1 R132H antibody-antigen binding.
Unfortunately, we cannot perform the test currently due to resource limitations of the fresh human tumor samples. Certainly, the follow-up tests will be performed in our future studies when these samples are available. Similar analyses may also be performed with the cell lines, BT142 mut/- and HepG2, and their xenograft tumors.
- Comments and/or suggestions of the reviewer:
Since the antibody concentration used in the dot blot (0.3 µg/ml) is lower than the concentration used for cell pellet analysis (0.64 µg/ml) and the concentration used for histological samples (2.54 µg/ml), please add a comment as to whether the cross-reactivity towards the R132S ( & R132G) mutant could be of concern for the IHC analysis.
Responses of authors:
First, the working concentrations of the IDH1 antibodies were determined by separate optimization tests for different protocols including Western blotting, dot immunoassay, the immunohistochemical tests on cell pellet block and on human tissue samples.
Second, it is generally understandable that solid-phase immunoreactions require higher antibody concentrations than the liquid-phase assays, like ELISA, and that, among the solid-phase reactions, immunohistochemistry on FFPE samples often require higher antibody concentration levels in comparison to dot immunoassays and IHC reactions on cell line samples. This is one of the reasons why the best-quality antibodies (with high affinity and low background) are needed for immunohistochemistry on routine tissue samples.
Third, it is interesting to note the specificity difference of MRQ-67 between ELISA and the solid-phase assays. With ELISA, it showed high-level binding to IDH1 R132S, and lower-level reactions to R132L, R132G and R132C. With the dot immunoassay, however, the cross-reactions were not demonstrated. At the moment, we have not fully understood the difference, but we tend to ascribe the phenomenon to the possible conformational changes of these peptides when mobilized on some matrix.
For our purpose, we would feel satisfied if an IHC assay could only be positive in samples with R132 mutants, but not with IDH1 wt or IDH2 wt. Until now, it is confirmed that MRQ-67 do not react with IDH1 wt and IDH2 wt, but we do not have any immunohistochemical evidence that MRQ-67 also recognize samples with IDH1 mutants other than R132H.
Theoretically, the IDH-mutant gliomas include all of the lesions carrying mutations in IDH1 or in IDH2, which are frequently affecting IDH1 R132 or IDH2 R172 codons. During the neuropathologic practice, around 90% of the IDH-mutant gliomas are with the IDH1 R132H mutation.
- Comments and/or suggestions of the reviewer:
Correct line 38: "sequencing if with sufficient..."
Responses of authors:
Lines 39-40 (Line 38 in the submitted version). “This could be achieved through sequencing if with sufficient and qualified samples.” is changed to “This could be achieved through sequencing if tumor samples are sufficient and qualified.”